# Hospital Mortality and Morbidity in Diabetic Patients with COVID-19: A Retrospective Analysis from the UAE

**DOI:** 10.3390/ijerph21060697

**Published:** 2024-05-29

**Authors:** Yehia S. Mohamed, Mamoun Mukhtar, Akrem Elmalti, Khalid Kheirallah, Debadatta Panigrahi, Eman Y. Abu-rish, Ibrahim Bani, Eiman Mohamed Nasor, Wafa Ahmed, Abdallah Alzoubi

**Affiliations:** 1Department of Pathological Sciences, College of Medicine, Ajman University, Ajman P.O. Box 346, United Arab Emirates; d.panigrahi@ajman.ac.ae (D.P.); i.bani@ajman.ac.ae (I.B.); 201810402@ajmanuni.ac.ae (E.M.N.); 201810396@ajmanuni.ac.ae (W.A.); a.alzoubi@ajman.ac.ae (A.A.); 2Department of Microbiology and Immunology, Faculty of Pharmacy (Boys), Al-Azhar University, Cairo 11562, Egypt; 3Rashid Center for Diabetes and Research, Sheikh Khalifa Medical City Ajman, Al Jurf, Ajman P.O. Box 5166, United Arab Emirates; mamoun.mukhtar@skmca.ae; 4Dr. Suliman Habib Hospital, UmmHurair2, Dubai 11372, United Arab Emirates; aelmalti@hotmail.com; 5Department of Public Health and Community Medicine, Faculty of Medicine, Jordan University of Science and Technology, Irbid 22110, Jordan; kkheiral@gmail.com; 6Department of Biopharmaceutics and Clinical Pharmacy, School of Pharmacy, The University of Jordan, Amman 11942, Jordan; e.aburish@ju.edu.jo

**Keywords:** diabetes, COVID-19, SARS-CoV-2, comorbidities, mortality, UAE

## Abstract

Background: Although we are four years into the pandemic, there is still conflicting evidence regarding the clinical outcomes of diabetic patients hospitalized with COVID-19. The primary objective of this study was to evaluate the in-hospital mortality and morbidity of diabetic versus nondiabetic patients hospitalized with COVID-19 in the Northern UAE Emirates. Methods: A retrospective analysis was performed on clinical data from patients with or without diabetes mellitus (DM) who were admitted to the isolation hospital with COVID-19 during the first and second waves of the disease (March 2020 to April 2021). The assessed endpoints were all-cause in-hospital mortality, length of hospitalization, intensive care unit (ICU) admission, and mechanical ventilation. Results: A total of 427 patients were included in the analysis, of whom 335 (78.5%) had DM. Compared to nondiabetics, diabetic COVID-19 patients had a significantly longer in-hospital stay (odds ratio (OR) = 2.35; 95% confidence interval (CI) = 1.19–4.62; *p* = 0.014), and a significantly higher frequency of ICU admission (OR = 4.50; 95% CI = 1.66–7.34; *p* = 0.002). The need for mechanical ventilation was not significantly different between the two groups (OR: distorted estimates; *p* = 0.996). Importantly, the overall in-hospital mortality was significantly higher among diabetic patients compared to their nondiabetic counterparts (OR = 2.26; 95% CI = 1.08–4.73; *p* = 0.03). Conclusion: DM was associated with a more arduous course of COVID-19, including a higher mortality rate, a longer overall hospital stay, and a higher frequency of ICU admission. Our results highlight the importance of DM control in COVID-19 patients to minimize the risk of detrimental clinical outcomes.

## 1. Introduction

The emergence of the Severe Acute Respiratory Syndrome Coronavirus 2 (SARS-CoV-2) pandemic in late 2019 has spurred an unprecedented surge in scientific research. People worldwide have faced both immediate and secondary health impacts from this crisis. Over 700 million COVID-19 cases have been officially recorded, with the global death toll surpassing 7 million [1]. The COVID-19 pandemic has intensified the challenges for individuals with diabetes mellitus (DM) and the healthcare system due to the interrelated social, healthcare, and economic repercussions associated with this chronic illness [2].

Recent studies have provided substantial evidence that patients with DM and metabolic dysfunction experience significantly worse clinical outcomes, including longer hospital stays, increased need for mechanical ventilation and ICU admissions, and higher mortality rates due to COVID-19 [3,4,5,6]. New data also suggest that COVID-19 could trigger new-onset DM and acute metabolic complications of pre-existing DM, such as hyperglycemia and diabetic ketoacidosis [7]. The exact mechanisms behind these connections remain unclear, but they likely involve the angiotensin-converting enzyme 2 (ACE2) receptor, which serves as a docking site for the SARS-CoV-2 virus and is present in key metabolic organs like the pancreas, particularly in β cells. SARS-CoV-2’s affinity for β cells can potentially damage these cells and impede insulin secretion, leading to hyperglycemia and ketoacidosis [7].

In June 2020, an international collaboration of diabetes experts launched a global registry called CoviDIAB to track cases of COVID-19-related DM. This registry aims to investigate the extent and progression of new-onset DM and metabolic dysfunction in COVID-19 patients, identify new causes, and determine the most effective treatments [8]. Understanding the complex interactions between DM and COVID-19 has become crucial for protecting and effectively treating individuals with DM or those at a heightened risk of metabolic dysfunction. With the rising prevalence of DM and other non-communicable diseases (NCDs) globally, including in the United Arab Emirates (UAE), it is imperative to prioritize the prevention and control of these health issues [9].

The generalizability of global findings on the relationship between COVID-19 and DM in the UAE’s patient population may be challenging due to differences in demographic characteristics, lifestyle, comorbidities, and ethnic diversity. This complexity poses a challenge in managing individuals with DM who also contract COVID-19 in the UAE. The UAE is a diverse country with a high net migration rate and a population composed of only 10% of UAE nationals. The country has a male-to-female ratio of 2.5, a predominantly South Asian population, a median age of 30.3 years, and a life expectancy of 77.7 years at birth [10]. The UAE government invests in healthcare innovation, particularly in digital technology and predictive health models, to enhance service delivery and public health outcomes. Notably, the UAE has reported the lowest COVID-19 fatality rate globally, with fewer than 2350 deaths [9]. However, the country faces significant health issues, particularly DM and cardiovascular diseases (CVDs), which are projected to exceed the global prevalence due to unhealthy lifestyles, obesity, and high smoking rates. However, the UAE is addressing these health challenges through preventive measures, health awareness initiatives, and early detection programs [9].

Therefore, investigating the correlation between DM and COVID-19-related morbidity and mortality, along with developing therapeutic strategies, will provide valuable insights for future research in these vulnerable groups. The primary aim of this retrospective study was to examine and analyze the clinical outcomes of individuals hospitalized with COVID-19, comparing those with and without DM.

## 2. Methods

### 2.1. Ethical Considerations

Ethical approvals were obtained from the Research Ethics Committee at Ajman University (M-H-F-Jun-7/2021), and the UAE’s Ministry of Health and Prevention (MOHAP) (MOHAP/DXB-REC/JJJ/No.52/2021). Participant informed-consenting was waived for this retrospective medical record analysis study. All procedures were conducted following the 1975 Helsinki Declaration and comparable ethical standards. A particular code was generated for each patient, and the file containing the link of the patient-specific code with their name and hospital file number was locked and password protected. Thus, the patients’ information confidentiality was guaranteed, and the data analysis was conducted on the de-identified database.

### 2.2. Study Design and Population

This study has retrospectively assessed the medical records of hospitalized COVID-19 patients at Sheikh Khalifa Medical City Ajman (SKMCA), who were admitted during the period from March 2020 to January 2021. Selected participants were adult patients with a confirmed COVID-19 diagnosis. Patients < 18 years old and pregnant women were excluded. The COVID-19 infection confirmation was revealed by reverse-transcriptase polymerase chain reaction (RT-PCR) detection of SARS-CoV-2 RNA in the nasopharyngeal swab.

### 2.3. Data Collection

Socio-demographic and clinical data were retrospectively extracted from the electronic medical records. Age, gender, nationality, employment status, weight, height, body mass index (BMI), smoking, and reported comorbidities were logged. BMI was categorized according to the standard World Health Organization classification into underweight (<18.5 kg/m^2^), normal weight (18.5–24.9 kg/m^2^), overweight (25.0–29.9 kg/m^2^), and obese (≥30.0 kg/m^2^). In addition, the baseline clinical status of the study participants at the time of admission was logged. This included presenting complaints, vital signs, radiological findings, and blood laboratory results (including glycemia, clotting, inflammatory, and organ dysfunction markers). Noteworthy, some clinical data were missing or suffering from overt inaccuracies or irregularities, such as the temperature and heart rate, and were thus omitted from the analysis. Additionally, the hospitalization details were logged, including admission date, received therapies, the highest level of care, end-result of admission, discharge/death date, and oxygen support details during hospitalization and their durations. DM was confirmed using the national criteria of the MOHAP [11]. Study participants were classified by DM diagnoses into DM group and non-DM group. For the DM group, diabetes-related features and indicators were reported, including type, treatments, and the associated complications. 

### 2.4. Study Primary and Secondary Outcomes

The primary evaluated outcome of this study was the all-cause in-hospital mortality. Secondary outcomes were patient morbidity as defined by the need for intensive care unit (ICU) admission, mechanical ventilation requirement, and the duration of hospitalization. 

### 2.5. Statistical Analysis

The IBM Statistical Package for the Social Sciences (SPSS), version 25.0, was used for data processing and analyses. Descriptive statistics, including frequencies and percentages, were calculated for the categorical variables, while continuous variables were presented using the mean (standard deviation, SD). Student t-test and Chi-square, as appropriate, were conducted to assess the differences in the study variables by DM status. The rate of all-cause mortality, the duration of hospitalization, the frequency of ICU admission, and the mechanical ventilation need were reported overall and linked to DM status. Odds ratios (OR) and their 95% confidence intervals (95% CI) were reported. Statistical significance was considered at a *p*-value of <0.05. With the provided sample size, using an alpha level of 0.5%, the power of the study to detect the difference between DM and non-DM patients for all clinical outcomes was more than 80%.

## 3. Results

### 3.1. Background Characteristics of the Study Population

Table 1 summarizes the baseline participants’ sociodemographic and clinical characteristics, overall and by DM status, and Appendix A details the DM-related features and indicators. The total number of enrolled patients was 427, of which 335 (78.5%) were diabetics. Approximately 98% of diabetic patients were of type 2 DM, and the majority (37.3%) were on oral antidiabetic medications (e.g., metformin and dapagliflozin) (Appendix A). The mean age of non-DM (47.8 ± 13.9 years) was not significantly different than DM patients (57 ± 13.1 years) (*p* = 0.35). The distribution of study participants by gender, nationality, smoking status, and BMI was not significantly different by DM status (Table 1). However, the two groups were significantly different by employment status, with the majority of nondiabetics being employed while the majority of diabetics were unemployed (*p* = 0.003). Comorbidities, such as cardiovascular diseases, dyslipidemia, and hypertension, were significantly higher among DM patients compared to non-DM patients (*p* = 0.001, 0.00, and 0.00, respectively). Other comorbidities, defined as diseases reported in <5 patients, such as asthma and hypo/hyperthyroidism, were also significantly higher among diabetic patients (*p* < 0.001). Furthermore, vital signs at the time of presentation revealed systolic blood pressure readings on the upper limit of normal (*p* = 0.12; non-DM vs. DM) and slightly elevated respiratory rates (*p* = 0.41; non-DM vs. DM). Though not clinically relevant, oxygen saturation at room air (SpO_2_) was significantly higher in the DM group (95.6 ± 4.6%) in comparison to the non-DM (92.8 ± 8.3%; *p* = 0.00), however, both levels were above the guideline-recommended cut-off point of hypoxia (90–94%) [12]. Blood glucose levels were, as expected, significantly higher in patients with DM (15.7 ± 6 mmol/L) than non-DM (7.2 ± 1.9 mmol/L, *p* = 0.00). However, both the WBC and CRP values were not significantly different between the two groups. Of clinical importance was the finding of significantly higher pro-calcitonin levels among DM when compared to non-DM (*p* = 0.03), albeit abnormal chest X-ray findings in the form of unilateral or bilateral patchy infiltrations were evident in the majority of both non-DM (88%) and DM patients (89%, *p* = 0.47), as detailed in Table 1.

Appendix A outlines the main symptoms reported by our patients at the time of COVID-19 diagnosis. Overall, dyspnea, fever, and cough were the most frequently reported presenting symptoms in the study population (68.1%, 63.5%, and 60.9%, respectively). Statistically significant differences between the two groups were only evident in the frequency of myalgia (38% in non-DM vs. 13% in DM, *p* = 0.00), vomiting (24% in non-DM vs. 9% in DM, *p* = 0.00), and diarrhea (18.5% in non-DM vs. 8.4% in DM, *p* = 0.006). 

### 3.2. COVID-19 Treatment Regimens

As shown in Table 2, antibacterial agents (e.g., ceftriaxone, azithromycin) and corticosteroids (dexamethasone) were administered to the majority of patients in our study (52.2% and 58.3%, respectively). Since this study reports on patients’ data during the early phases of the COVID-19 pandemic, it was not surprising to find a limited use of antivirals (e.g., remdesivir) and immunoglobulins in our cohort of patients (19.2% and 1.4%, respectively). A combination of two agents (frequently an antibiotic and a corticosteroid) was found in 32% of patients, while the addition of a third agent (remdesivir in particular) was seen in only 8.9% of patients. On a closer analysis of the therapeutic regimens utilized in our study population, we interestingly found a significantly higher frequency of using corticosteroids among nondiabetics (100.0% vs. 46.9% in diabetics; *p* = 0.00). The same pattern applied to the use of the 2-drug (38.0% vs. 30.1% in diabetics; *p* = 0.00) and the 3-drug combination regimens (13.0% vs. 7.8% in diabetics; *p* = 0.00).

### 3.3. Clinical Outcomes of COVID-19 Infection

Distribution of the primary and secondary clinical outcomes of COVID-19 infection among the study participants by DM status is presented in Table 3. Overall, we found a significantly higher overall in-hospital mortality among diabetic patients (19.7% vs. 9.8% in nondiabetic patients; *p* = 0.03). The mean total duration of hospitalization in the study population was approximately 13 ± 12 days. Diabetic COVID-19 patients had a significantly longer stay in the hospital in comparison to nondiabetic patients (13.9 ± 13.1 vs. 9.4 ± 8.3 days; *p* = 0.00). Further, the frequency of admission to the intensive care unit (ICU) was significantly higher among diabetics when compared to nondiabetic patients (24.2% vs. 12.0%, respectively; *p* = 0.01). The need for invasive mechanical ventilation, as well, was remarkably higher in diabetic COVID-19 patients in comparison to nondiabetic patients (21.2% vs. 0.0%, respectively; *p* = 0.00). 

### 3.4. The Association between DM Status and Clinical Outcomes

To further assess the relationship between DM status and the clinical outcomes of the study (namely in-hospital mortality, ICU admission, duration of hospitalization, and mechanical ventilation), we estimated the odds ratio (OR) of each clinical outcome by DM status, as indicated in Table 4. Compared to their non-DM counterparts, DM COVID-19 patients were at a higher risk of in-hospital mortality (OR = 2.263; 95% C.I: 1.081–4.736; *p* = 0.03) and needed significantly longer durations of hospitalization (OR = 4.503; 95% C.I: 1.664–7.341; *p* = 0.002) and ICU admission (OR = 2.348; 95% C.I: 1.192–4.624; *p* = 0.014). However, DM was not significantly associated with the need for mechanical ventilation in this sample of patients (*p* = 0.996), as shown in Table 4. 

## 4. Discussion

This study expanded on a limited number of reports from the UAE to assess and contrast the demographic and clinical features, along with the outcomes, of COVID-19 infection in DM patients in comparison to non-DM patients. We report here that DM is associated with detrimental COVID-19 outcomes, including higher in-hospital mortality, longer overall hospital stays, and a higher frequency of ICU admission.

DM is a prominent contributor to morbidity and mortality on a global scale [13]. This condition is characterized by a hyperglycemic inflammatory state that results in impaired host immune responses and increased susceptibility to infections, including COVID-19 [14]. In line with our results, a recent meta-analysis indicated a significant increase in the severity of COVID-19 infection among individuals with diabetes, with a two to three-fold rise [15]. Another study revealed that diabetic patients exhibited higher rates of unfavorable clinical outcomes and mortality [6]. Moreover, Hussain et al. [16] have shown that the death rate of COVID-19 was 9.3% in non-DM patients, 24.96% in patients with newly diagnosed DM, and 16.03% in patients with pre-existing DM. In another study by Karagiannidis et al. [17], it was shown that 38.9% of individuals with DM who were infected with COVID-19 in Germany required ventilation. Additionally, a survey conducted in England involving 23,804 COVID-19 patients showed that 32% of patients with type 2 DM and 1.5% with type 1 DM succumbed to the disease [18]. 

Numerous studies have demonstrated a correlation between DM and a heightened mortality risk ranging from 22% to 31% concerning COVID-19 infection, in contrast to individuals without DM with a mortality risk ranging from 2% to 4% [19]. According to the Chinese Center for Disease Control and Prevention [20], the COVID-19 mortality rate among those with DM is 7.3%, which is higher than the overall mortality rate of 2.3%. In a study conducted by Barron et al., a total of 23,804 fatalities related to COVID-19 were documented. Among these cases, about one-third of the patients were found to have DM, with 31.4% having type 2 DM and 1.5% having type 1 DM. According to this study, the odds ratio for in-hospital fatalities associated with COVID-19 was shown to be 3.51 for individuals with type 1 DM and 2.03 for individuals with type 2 DM, in comparison to the DM-free population [3].

Our study revealed that DM patients experienced severe COVID-19 consequences resulting in a higher rate of ICU admission (24% vs. 12% in nondiabetics, *p* < 0.01) and a longer overall hospital stay (13.94 days vs. 9.43 days in nondiabetics, *p* < 0.001). Such advanced and grave progression of COVID-19 in diabetic patients might explain the significantly higher in-hospital mortality rates (19.7%) compared to nondiabetic patients (9.8%, *p* = 0.03). The findings of our study align with a previous study, which has shown a correlation between DM and increased rates of ICU hospitalizations, invasive ventilation utilization, and longer hospital stays [14,21]. In the UAE, a limited number of studies tackled the consequences of COVID-19 among diabetic patients with varying results. Elemam et al. reported in 2021 [22] that COVID-19 patients with DM had a higher percentage of critical illness and ICU admissions. Mortality rates were estimated to be 36% in diabetic versus 10% in nondiabetic patients. The number of diabetic patients in this particular study was considerably less than ours (*N* = 111 vs. 335 in our study). In another study by Alkhemeiri et al. in 2022 [23], diabetic COVID-19 patients had a higher frequency of ICU admission and intubation compared to nondiabetics. The mortality rate in diabetic patients was 5.9%. The sample size of diabetic patients included in this study was comparable to ours, however, the study reported on a different subpopulation in the UAE (Sothern Emirates) and timeframe (up to April 2021), where COVID-19 vaccination was mandated by the country.

Intriguingly, we found that procalcitonin levels were significantly higher among DM when compared to non-DM patients (*p* = 0.03), albeit abnormal chest X-ray findings in the form of unilateral or bilateral patchy infiltrations were evident in the majority of both non-DM (88%) and DM patients (89%, *p* = 0.47). Historically, the diagnostic value of procalcitonin as a marker of bacterial infection in children and adults has been a topic of debate [24,25]. High procalcitonin levels in COVID-19 patients have been reported in several observational clinical studies. For instance, Huang et al. demonstrated in 2020 that high procalcitonin levels may be an indicator of disease severity in COVID-19 [25]. Xu et al. (2022) have further extended the clinical utility of procalcitonin into guiding antimicrobial stewardship programs and whether COVID-19 patients should receive antibiotic therapy [26]. However, other recent studies provided conflicting evidence on the diagnostic value of procalcitonin in COVID-19 patients. A study by Fisler et al. in 2023 revealed a low positive predictive value of procalcitonin in identifying either culture-confirmed or clinically suspected bacterial infection in children hospitalized with COVID-19 [27]. Additionally, Tao et al. concluded in 2024 that procalcitonin levels correlated with the risk of hyperglycemic crisis in DM patients with COVID-19 but not with a poor clinical outcome [28]. Further studies are warranted to clarify the clinical utility of procalcitonin in COVID-19 patients.

Additionally, previous studies have indicated a frequent coexistence of DM and hypertension [29]. Our results revealed that the majority of the diabetic cohort had prior hypertension (63%) and other cardiovascular diseases (29%), which is significantly higher compared to the nondiabetic group. These findings align with worldwide studies that have confirmed the frequency of DM and hypertension as prevalent comorbidities in COVID-19 cases [4]. Sourij and colleagues [5] reported similar observations, with hypertension being present prior to COVID-19 in 77% of the diabetic individuals. Riddle et al. [29] demonstrated that 56.9% of COVID-19 patients with DM had prior hypertension. Conversely, hypertension was identified as the most common coexisting condition observed in patients with COVID-19 [30], and Elemam et al. [22] highlighted the correlation between hypertension and the severity of illness in COVID-19 patients. While our study did not specifically address the management of hypertension in COVID-19, it is crucial to highlight the evidence supporting the positive impact of continuing ACE/ARB medication for hypertension during COVID-19 infection, as it improves patients’ clinical outcomes [31]. 

Regarding the management and treatment of patients with COVID-19, our analysis showed that both groups (diabetics and nondiabetics) almost required the same therapeutic interventions apart from corticosteroids, which were required more frequently in the non-diabetic group (100%) compared to the diabetic cohort (47%; *p* < 0.001). This could be attributed to the fact that glucocorticoids inherently worsen glycemic control and result in hyperglycemia. The same pattern applied to the use of combination therapy as it was higher in non-DM than in DM patients, given that corticosteroids were the main agent to be added in two or three-drug regimens. It must be noted, however, that our results reflect the management approaches during the early times of the COVID-19 pandemic, where some agents were yet to be approved for use (e.g., remdesivir) and treatment protocols were quite inconsistent and left for the physician’s best clinical judgment. 

Despite the importance of our findings, this study has several limitations. First, the retrospective nature of this study and being a single-center study perhaps impede the generalization of our findings. Second, some clinical data were missing or suffering from overt inaccuracies or irregularities, such as the temperature and heart rate, and were thus omitted from analysis. Third, the sample size in both groups was relatively small. However, the comparative cohort design of the study examining diabetic vs. nondiabetic patients with COVID-19 could strengthen the level of evidence provided. Fourth, since our patients’ records were during the first and second wave of the pandemic we could not study the effect of vaccination on the disease outcomes. However, other studies from the UAE revealed that diabetic participants who did not receive the initial dose of the vaccine had almost six times higher odds of requiring non-invasive ventilation compared to diabetic subjects who received the first dose of the vaccine, and individuals who were diabetic and did not receive the initial dose of the vaccine had almost six times higher likelihood of being admitted to the ICU compared to individuals who were diabetic but received the first dose of the vaccine [23]. Fifth, we have not aimed to monitor the prognosis of hyperglycemia after patients’ recovery. However, prior research conducted by Cromer et al. [32] and Laurenzi et al. [33] provided evidence that glycemic control improves and DM goes into remission once COVID-19 infection is resolved, especially with newly diagnosed patients with DM. Rezel-Potts and colleagues [34] also verified the likelihood of developing DM after contracting COVID-19, with the risk persisting for a minimum of 12 weeks before gradually decreasing.

## 5. Conclusions

In conclusion, our study reports a more severe clinical picture and serious negative outcomes of COVID-19 in diabetic compared to nondiabetic patients during the early waves of the pandemic in the UAE. Overall, our results fall in line with the globally reported findings on this interplay of COVID-19 and DM. Given the high prevalence of DM and comorbid cardiovascular diseases and hypertension in the UAE community, our analysis may have identified implications for improving the management protocols for diabetic COVID-19 patients. However, larger randomized and goal-oriented clinical trials are warranted to validate our findings. This, however, will be particularly challenging as the acute and overwhelming phases of the pandemic are over. We recommend conducting thorough and extended monitoring of patients with DM and those who develop DM after contracting COVID-19 to prevent their deterioration and improve their clinical outcomes.

## Figures and Tables

**Table 1 ijerph-21-00697-t001:** Distribution of background sociodemographic and clinical patient characteristics at the time of admission by DM status (*N* = 427).

			DM Status		
Item		Total (*N* = 427)	Nondiabetic (*n* = 92)	Diabetic (*n* = 335)	*p*-Value
**Age (years)** *Mean (SD)* *Median (IQR)*		55.57 (13.88) 56 (19)	47.76 (13.97) 48 (20)	57.72 (13.07) 59 (17)	0.35
**Gender** *n* (%)	** *Male* **	293 (68.6)	66 (71.7)	227 (67.8)	0.53
	** *Female* **	134 (31.4)	26 (28.3)	108 (32.2)	
**Nationality** *n* (%)	** *Emirati* **	66 (15.5)	8 (8.7)	58 (17.3)	0.05
	** *Non-Emirati* **	361 (84.5)	84 (91.3)	277 (82.7)	
**Employment** *n* (%)	** *Employed* **	215 (50.4)	59 (64.1)	156 (46.6)	0.003
	** *Unemployed* **	212 (49.6)	33 (35.9)	179 (53.4)	
**BMI** *Mean (SD)*		29.71 (6.6)	29.66 (5.84)	29.73 (6.81)	0.27
**Smoking** *n* (%)	** *Smoker* **	31 (7.3)	10 (10.9)	21 (6.3)	0.17
	** *Nonsmoker* **	396 (92.7)	82 (89.1)	314 (93.7)	
**Comorbidities** *n* (%)	** *CVD* **	108 (25.3)	11 (12.0)	97 (29.0)	<0.001
	** *Dyslipidemia* **	118 (27.6)	10 (10.9)	108 (32.2)	<0.001
	** *HTN* **	235 (55.0)	23 (25.0)	212 (63.3)	<0.001
	** *Other ** **	183 (42.9)	24 (26.1)	159 (47.5)	<0.001
**Vital signs** *Mean (SD)*	** *RR* **	21.24 (4.34)	21.29 (4.62)	21.23 (4.26)	0.41
	** *SBP* **	132.11 (19.48)	131.01 (16.54)	132.42 (20.23)	0.12
	** *DBP* **	77.86 (12.57)	80.17 (11.54)	77.23 (12.79)	0.31
	** *SpO_2_* **	94.92 (5.75)	92.75 (8.32)	95.54 (4.60)	0.00
**Laboratory findings** *Mean (SD)*	** *Blood Glucose* **	13.90 (6.43)	7.21 (1.98)	15.72 (6.00)	0.00
	** *WBC* **	7.68 (5.59)	7.82 (4.69)	7.64 (5.82)	0.57
	** *CRP* **	103.47 (89.74)	105.34 (87.06)	102.97 (90.57)	0.76
	** *Pro-Calcitonin* **	0.89 (3.47)	0.49 (1.02)	1.01 (3.86)	0.03
**Abnormal CXR** ** *n* (%)		379 (88.8)	81 (88.0)	298 (89.0)	0.47

*; Other comorbidities were defined as diseases reported among <5 patients, including asthma, hypothyroidism, thyrotoxicosis, epilepsy, nephrolithiasis, glaucoma, Alzheimer’s disease, and benign prostatic hyperplasia. **; Abnormal chest X-ray was defined as the finding of unilateral or bilateral patchy infiltrations. DM: diabetes mellitus; IQR: interquartile range; BMI: body mass index; CVD: cardiovascular diseases; HTN: hypertension; CXR: chest X-ray; SD: standard deviation.

**Table 2 ijerph-21-00697-t002:** COVID-19 treatment regimen during hospital admission in nondiabetic and diabetic patients.

Item		Total (*N* = 427)	Nondiabetic (*n* = 92)	Diabetic (*n* = 335)	*p*-Value
**Antivirals** *n* (%)		82 (19.2)	15 (16.3)	67 (20.0)	0.46
**Antibiotics** *n* (%)		223 (52.2)	42 (45.7)	181 (54.0)	0.16
**Corticosteroids** *n* (%)		249 (58.3)	92 (100.0)	157 (46.9)	<0.001
**Immunoglobulins** *n* (%)		6 (1.4)	2 (2.2)	4 (1.2)	0.61
**Combination Therapy** *n* (%)	2 Drugs 3 Drugs	136 (31.9) 38 (8.9)	35 (38.0) 12 (13.0)	101 (30.1) 26 (7.8)	<0.001

**Table 3 ijerph-21-00697-t003:** Distribution of the primary and secondary clinical outcomes, overall and by DM status.

Outcome	Total (*N* = 427)	Nondiabetic (*n* = 92)	Diabetic (*n* = 335)	*p*-Value
**Overall in-hospital mortality** *n* (%)	75 (17.6)	9 (9.8)	66 (19.7)	0.03
**Total duration of hospitalization in days** *Mean (SD)* *Median (IQR)*	12.97 (12.39) 9 (11)	9.43 (8.30) 7 (6)	13.94 (13.14) 10 (12)	<0.001
**Admission to ICU** *n* (%)	92 (21.5)	11 (12.0)	81 (24.2)	0.01
**Mechanical ventilation** *n* (%)	71 (16.6)	0 (0.0)	71 (21.2)	<0.001

DM: diabetes mellitus; IQR: interquartile range; ICU: intensive care unit; SD: standard deviation.

**Table 4 ijerph-21-00697-t004:** Effect of DM status on the primary and secondary clinical outcomes under investigation.

Outcome	DM Status	Odds Ratio (OR)	95% Confidence Interval (Min.–Max.)	*p*-Value
**Overall in-hospital mortality**	** Non-DM **	REF	1.081–4.736	0.03
	** DM **	2.263		
**Total duration of hospitalization**	** Non-DM **	REF	1.192–4.624	0.014
	** DM **	2.348		
**Admission to ICU**	** Non-DM **	REF	1.664–7.341	0.002
	** DM **	4.503		
**Mechanical ventilation**	** Non-DM **	REF	Distorted estimates	0.996
	** DM **	Distorted estimates		

DM: diabetes mellitus; ICU: intensive care unit; REF: reference; Min.: minimum; Max.: maximum.

## Data Availability

Data are available upon request.

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
