# Peer review of "Hospital Mortality and Morbidity in Diabetic Patients with COVID-19: A Retrospective Analysis from the UAE"

_ijerph, 2024, doi:10.3390/ijerph21060697_

Round 1
Reviewer 1 Report
Comments and Suggestions for Authors
1. The introduction, from my point of view, is not comprehensive and perhaps requires to consider more information on the local figures of diabetes and COVID-19 in UAE for better understanding.
2. Several abbreviations were not defined in the manuscript, such as SKMCA (line 87), MOHAP and T2d (104).
3. Please unify the abbreviation of Diabetes mellites: (T2D) and (DM).
4. The allocated time of study for collecting data was less than a year, why?
5. They must review the results. Line 122, the result did not report the sig. changes according to Employment.
6. Table 1, The authors reported only the CVS comorbidities (CVS HT and dyslipdemia), meanwhile very important and relevant diseases are missed such as respiratory, immune, malignancy and renal. I think either add more comorbidities or specify the current data as CVS comorbidities with further justification for this point.
7. Moreover, atherosclerosis is not presented in the Table 1.
8. Table 3 looks ambiguous since many COVID-19 symptoms were missed. According to WHO the symptoms of Covid 19 included:
muscle aches and heavy arms or legs
severe fatigue or tiredness
runny or blocked nose, or sneezing
headache
sore eyes
dizziness
tight chest or chest pain
hoarse voice
numbness or tingling
appetite loss,
abdominal pain or diarrhoea
difficulty sleeping.
https://www.who.int/news-room/fact-sheets/detail/coronavirus-disease-(covid-19)
9. The same issue in the vital signs: temperature and heart rate were not reported.
10. What did they mean by abnormal CXR and how? Kindly indicate the clinical abnormalities in CXR.
11. The presentation of results in the text should be aligned with the same sequence of Table: Table 5 (Line 180) and Table 6 (Line 196).
12. The title of Tables 5 and 6 can be changed to represent the type of test used (association or prediction).
13. Plz add the abbreviations of Table 6.
14. I would suggest adding figures to the manuscript.
15. In the discussion (line 210), the authors stated that (We report here that diabetes is associated 208 with detrimental COVID-19 outcomes, including higher in-hospital mortality, longer 209 overall hospital stays, and higher frequency of mechanical ventilation and ICU admission.), while as appear in Table 6 DM was not significantly associated with the need for Mechanical ventilation.
16. The authors did not discuss the significant the significant change in Spo2 and Pro-Calcitonin.
17. The Conclusion section must be specific & concrete in relation to his objectives.
“Enhancing the care and management of diabetes and the associate complications during COVID-19 infection …”, did you evaluate the healthcare system?
“However, larger randomized and goal-oriented clinical trials are warranted to validate our findings, particularly as the acute and overwhelming phases of the pandemic are over”. How can you recruit new COVID-19 in-patients to confirm the findings nowadays?
Addressing these points will significantly enhance the manuscript in terms of clarity, coherence, and scientific contribution. Your study has the potential to make a meaningful impact in the field, and these improvements will help in achieving that.
Comments on the Quality of English LanguageThe grammatical errors with a paraphrase of the sentence and abbreviations must be ensured before the final publication. Thus, comprehensive revision and editing are needed before accepting the manuscript
Author Response
- The introduction, from my point of view, is not comprehensive and perhaps requires to consider more information on the local figures of diabetes and COVID-19 in UAE for better understanding.
Response: Thank you for this helpful suggestion: A new paragraph has been added in the Introduction section for an explanation of the local figures on diabetes and COVID-19 in the UAE.
- Several abbreviations were not defined in the manuscript, such as SKMCA (line 87), MOHAP, and T2d (104).
Response: Thank you for this helpful comment. This has been amended and highlighted throughout the manuscript.
- Please unify the abbreviation of Diabetes mellites: (T2D) and (DM).
Response: Thank you for this helpful suggestion. This has been amended and highlighted throughout the manuscript.
- The allocated time of study for collecting data was less than a year, why?
Response: Thank you for this comment. The patient data for the first reported waves of the pandemic, March 2021 to January 2022, represent one of the highest peaks of the disease before the COVID-19 vaccine introduction in the UAE. In addition, the ethical approval was issued for only one year starting July 2021, and due to the logistic challenges during the pandemic, such as restricted access to the hospital, shortage of time of the collaborator clinicians, and the home-isolation procedures in case of any suspicious signs on the volunteering students, we haven’t been able to collect more records during the allocated time. This is the reason behind the low number of patients and the short duration of data collection.
- They must review the results. Line 122, the result did not report the sig. changes according to Employment.
Response: Thank you for this valuable observation. This has been amended and highlighted in the Results section.
- Table 1, The authors reported only the CVS comorbidities (CVS HT and dyslipidemia), meanwhile very important and relevant diseases are missed such as respiratory, immune, malignancy and renal. I think either add more comorbidities or specify the current data as CVS comorbidities with further justification for this point.
Response: Thank you for this insightful comment. We agree with the Reviewer regarding the important and relevant comorbidities in COVID-19 outcomes. We are reporting on what was found in the medical records of our patient cohort, and have actually included in Table 1 a category of “Other comorbidities,” which was defined as diseases reported in <5 patients. The text and Table 1 footnote have been amended and highlighted to indicate this.
- Moreover, atherosclerosis is not presented in the Table 1.
Response: Thank you for this helpful comment. The term atherosclerosis has been deleted.
- Table 3 looks ambiguous since many COVID-19 symptoms were missed. According to WHO the symptoms of Covid 19 included:
Response: Thank you for this insightful comment. We agree with the Reviewer regarding the important and relevant presenting symptoms of COVID-19. However, we are reporting on what was found in the medical records of our patient cohort.
- The same issue in the vital signs: temperature and heart rate were not reported.
Response: Thank you for this important observation. We admit that some of the clinical variables have been missing or suffering from significant inaccuracies or irregularities in the hospital records, such as the logs for temperature and heart rate. Thus, these variables have been omitted from the final analysis. We have added a comment to this effect in the Methods section and have also identified this as a limitation in the Discussion section.
- What did they mean by abnormal CXR and how? Kindly indicate the clinical abnormalities in CXR.
Response: Thank you for this significant comment. We have explained the term “abnormal CXR: Abnormal chest X-ray was defined as the finding of unilateral or bilateral Abnormal patchy infiltration” in the footnote of Table 1.
- The presentation of results in the text should be aligned with the same sequence of Table: Table 5 (Line 180) and Table 6 (Line 196).
Response: Thank you for this insightful observation. This has been amended in the text as per your suggestion.
- The title of Tables 5 and 6 can be changed to represent the type of test used (association or prediction).
Response: Thank you for this important comment. For Table 5 (now Table 3), it does present the distribution of study participants by the clinical outcomes under investigation. This reflects an association as per the provided p-value. Thus, the title has been changed to: “Distribution of the primary and secondary clinical outcomes, overall and by DM status.”
As for Table 6 (now Table 4), it does reflect the effect of DM on clinical indicators. This is a clear association and it is already reflected in the term “Effect”. Thus, we have kept the title as it is hoping our assumption satisfies the Reviewer.
- Plz add the abbreviations of Table 6.
Response: Thank you for this helpful comment. This has been added.
- I would suggest adding figures to the manuscript.
Response: Thank you for this valid suggestion. We do believe adding a figure could support our result presentations, but we believe also that our major outcomes and associations are sufficiently presented with tables. Please, if you have a specific figure to suggest, we can add it.
- In the discussion (line 210), the authors stated that (We report here that diabetes is associated 208 with detrimental COVID-19 outcomes, including higher in-hospital mortality, longer overall hospital stays, and higher frequency of mechanical ventilation and ICU admission.), while as appear in Table 6 DM was not significantly associated with the need for Mechanical ventilation.
Response: Thank you for this helpful comment. This has been amended and highlighted in the Discussion section.
- The authors did not discuss the significant change in SpO2 and Pro-Calcitonin.
Response: Thank you for this insightful recommendation. A brief discussion of these results has been added and highlighted in the Discussion section.
- The Conclusion section must be specific & concrete in relation to his objectives. “Enhancing the care and management of diabetes and the associated complications during COVID-19 infection …”, did you evaluate the healthcare system?
Response: Thank you for this insightful recommendation. This has been amended and highlighted in the Conclusion section.
- “However, larger randomized and goal-oriented clinical trials are warranted to validate our findings, particularly as the acute and overwhelming phases of the pandemic are over”. How can you recruit new COVID-19 in-patients to confirm the findings nowadays?
Response: Thank you for this insightful recommendation. This has been amended and highlighted in the Conclusion section.
Addressing these points will significantly enhance the manuscript in terms of clarity, coherence, and scientific contribution. Your study has the potential to make a meaningful impact in the field, and these improvements will help in achieving that.
Response: Thank you for your positive impression of our study, and we hope that the resubmitted revised form of the manuscript meets your satisfaction.
Reviewer 2 Report
Comments and Suggestions for Authors
This study aimed to investigate the impact of diabetes on 4 COVID-19 outcomes (mortality, length of stay, ICU admission, and mechanical ventilation).
This topic has been studied tens, if not hundreds, of times with several meta-analyses covering the same issue. The originality of this study is based on the lack of data from the UAE.
I recommend the following
1: Abstract: Use ORs and 95% CIs rather than numbers and percentages.
2: Introduction:
2:1: Previous studies should be cited.
2:2: Lines 56-64 need a reference.
2:3: Further details are needed to show the originality and justify conducting this study. Why do you think this study is needed?
2:4: The prevalence and burden of DM in the UAE should be discussed.
2:5: The authors mentioned the multi-ethnic and sociodemographic factors uniquely characterizing the UAE would make it difficult to use other results from different countries. However, no data were collected about the sociodemographic and ethnic characteristics of the included patients. So, this argument is incorrect and this limitation should be highlighted.
3: Methods:
3:1: How did the authors select their subjects?
3:2: Did the authors calculate the minimum sample size required?
4: Results:
The results need to be rearranged. Since the authors claim investigating the impact of DM on the 4 major outcomes is the purpose of the study, many changes are suggested.
First, Table 1 should compare the sociodemographic and clinical characteristics of patients with and without DM. Thus, vital signs, lab findings, and X-rays in Table 3 should be moved to Table 1. DM-related features in Table 2 should be moved to Table 1 (one column only).
Second, the differences in medications in Table 2 and clinical presentations in Table 3 between patients with and without DM are not directly related to this study. So, the authors may move them to a supplementary file (Supplemental Tables 1 and 2).
Third: Tables 5 and 6 should be merged into one table (Now it will be Table 2). Importantly, it is not enough to calculate unadjusted ORs. In observational studies, the possibility of confounders is high; therefore, the authors should adjust their results for confounders; at least age, sex, and variables were shown to be statistically different. The authors may use 3 models (unadjusted, age-and-sex-adjusted, and multi-variable adjusted).
Comments on the Quality of English Language
No major issues were detected.
Author Response
1: Abstract: Use ORs and 95% CIs rather than numbers and percentages.
Response: Thank you for this helpful comment. This has been amended and highlighted in the Abstract (Lines: 26-33).
2: Introduction:
2:1: Previous studies should be cited.
Response: Thank you for this helpful reminder. Citations have been added and highlighted.
2:2: Lines 56-64 need a reference.
Response: Thank you for this helpful reminder. Two references have been added and highlighted.
2:3: Further details are needed to show the originality and justify conducting this study. Why do you think this study is needed?
Response: Thank you for this helpful comment. A new paragraph about the unique demography of the UAE. Although many studies currently are available, there is still a need for further elucidation on the co-morbidities and mortalities associated with diabetes mellitus in the UAE’s community.
2:4: The prevalence and burden of DM in the UAE should be discussed.
Response: Thank you for this helpful comment. A new paragraph about this point has been added and highlighted in the Introduction section.
2:5: The authors mentioned the multi-ethnic and sociodemographic factors uniquely characterizing the UAE would make it difficult to use other results from different countries. However, no data were collected about the sociodemographic and ethnic characteristics of the included patients. So, this argument is incorrect and this limitation should be highlighted.
Response: Thank you for highlighting this point. The newly added paragraph in the introduction delves deeper into the demographic nature of the UAE, with Table 1's presented demographic data representing the study participants based on their ethnic origin. This data, fortunately, aligns with the average percentage of the UAE's current population.
3: Methods:
3:1: How did the authors select their subjects?
Response: Thank you for your inquiry. The study has retrospectively assessed the medical records of hospitalized COVID-19 patients at Sheikh Khalifa Medical City Ajman (SKMCA), who were admitted in the first few waves of the pandemic from March 2020 to January 2021. Selected participants were adult patients with confirmed COVID-19 molecular testing. Records of patients <18 years old and/or pregnant women were excluded.
3:2: Did the authors calculate the minimum sample size required?
Response: Thank you for this important comment. For the sample size calculation, please note that we used secondary data and did not prospectively collect data to conduct the research. We, however, estimated the power to detect a difference between DM and non-DM groups and for all comparisons, the estimated power was above 80%. This is the normal practice usually used when a secondary dataset is used. In the methods section, we have added the following under stat analysis: “With the provided sample size, using an alpha level of 0.5%, the power of the study to detect a difference between DM and non-DM patients for all clinical outcomes was more than 80%.”
4: Results:
The results need to be rearranged. Since the authors claim investigating the impact of DM on the 4 major outcomes is the purpose of the study, many changes are suggested. First, Table 1 should compare the sociodemographic and clinical characteristics of patients with and without DM. Thus, vital signs, lab findings, and X-rays in Table 3 should be moved to Table 1. DM-related features in Table 2 should be moved to Table 1 (one column only). Second, the differences in medications in Table 2 and clinical presentations in Table 3 between patients with and without DM are not directly related to this study. So, the authors may move them to a supplementary file (Supplemental Tables 1 and 2). Third: Tables 5 and 6 should be merged into one table (Now it will be Table 2).
Response: Thank you for your valuable suggestion. All Tables have been amended and highlighted throughout the Results section according to your suggestion.
Importantly, it is not enough to calculate unadjusted ORs. In observational studies, the possibility of confounders is high; therefore, the authors should adjust their results for confounders; at least age, sex, and variables were shown to be statistically different. The authors may use 3 models (unadjusted, age-and-sex-adjusted, and multivariable-adjusted).
Response: Thank you for these insightful comments. We appreciate the opportunity to address these concerns.
- While combining Tables 5 and 6 is interesting and provides an added value for someone with knowledge in statistics, it may get confusing to those with limited statistical background. We believe that estimating proportions and providing effects (ORs) are better separated to reduce confusion among readers and make the statistical analyses easily presented.
- As for the regression analyses, we agree that adjusting for potential confounding variables is crucial in many epidemiological studies to account for any factors that may influence the outcome of interest. However, in our specific study focusing solely on comparing the clinical outcomes of COVID-19 between diabetic and non-diabetic patients, we believe that the need for adjusting the odds ratio is mitigated because our study design focuses on a straightforward comparison between two groups (diabetic and non-diabetic patients). Please also note that for a regression model to be set, we need to identify one outcome variable first and then assess the relationship between a cluster of variables to that particular outcome variable. This means univariate, bivariate, and regression analyses.
Reviewer 3 Report
Comments and Suggestions for Authors
During the three former waves of SARS-Co-2 pandemic, mortality or Intensive Care Unit (ICU) admission of hypoxemic Covid-19 pneumonia in patients with diabetes mellitus (DM) have been reported up to 3 times more frequently than in patients with no DM [1, 2, 3]. So, this paper, even though not original, is an interesting contribution to previous articles in the same field.
- 1- Lim S, Bae J.H, Kwon H.S, Nauck M.A. Covid-19 and diabetes mellitus: from pathophysiology to clinical management. Nature Reviews 2021; 17: 11-30
- 2- Zhu L, She Z.G, Cheng X, et al. Association of blood glucose control and outcomes in patients with Covid-19 and pre-existing Type 2 diabetes. Cell Metab 2020; 31: 1068-1077 doi: 10.1016/j.cmet.2020.04.021
- 3- Mauvais-Jarvis Franck. Aging, male sex, obesity, and metabolic inflammation create the perfect storm for Covid-19. Diabetes 2020; 69 :1857-63
Study limitations are clearly described by authors.
Author Response
Thank you so much for the reviewer comment on our study, this is much appreciated.
Round 2
Reviewer 1 Report
Comments and Suggestions for Authors
I am pleased to acknowledge the author's response to all of my queries.
Thank you
Reviewer 2 Report
Comments and Suggestions for Authors
No more comments.